# Autoantibodies targeting GPCRs and RAS-related molecules associate with COVID-19 severity

Otavio Cabral-Marques [1,2,3,25] ✉, Gilad Halpert[4,5,25], Lena F. Schimke [1,25], Yuri Ostrinski [4,5,6], Aristo Vojdani[7,8], Gabriela Crispim Baiocchi [1], Paula Paccielli Freire [1], Igor Salerno Filgueiras [1], Israel Zyskind [9,10], Miriam T. Lattin[11], Florian Tran [12], Stefan Schreiber[12], Alexandre H. C. Marques [1], Desirée Rodrigues Plaça[2], Dennyson Leandro M. Fonseca[2], Jens Y. Humrich [13], Antje Müller[13], Lasse M. Giil [14], Hanna Graßhoff[13], Anja Schumann[13], Alexander Hackel[13], Juliane Junker[15], Carlotta Meyer[15], Hans D. Ochs[16], Yael Bublil Lavi [17], Carmen Scheibenbogen[18], Ralf Dechend[19], Igor Jurisica [20,21], Kai Schulze-Forster[15], Jonathan I. Silverberg[22], Howard Amital[4,17,23], Jason Zimmerman[10], Harry Heidecke[15], Avi Z. Rosenberg [24], Gabriela Riemekasten [13,25] ✉ & Yehuda Shoenfeld [4,5,6,25] ✉

COVID-19 shares the feature of autoantibody production with systemic autoimmune diseases. In order to understand the role of these immune globulins in the pathogenesis of the disease, it is important to explore the autoantibody spectra. Here we show, by a cross-sectional study of 246 individuals, that autoantibodies targeting G protein-coupled receptors (GPCR) and RAS-related molecules associate with the clinical severity of COVID-19. Patients with moderate and severe disease are characterized by higher autoantibody levels than healthy controls and those with mild COVID-19 disease. Among the anti-GPCR auto-antibodies, machine learning classification identifies the chemokine receptor CXCR3 and the RAS-related molecule AGTR1 as targets for antibodies with the strongest association to disease severity. Besides antibody levels, autoantibody network signatures are also changing in patients with intermediate or high disease severity. Although our current and previous studies identify anti-GPCR antibodies as natural components of human biology, their production is deregulated in COVID-19 and their level and pattern alterations might predict COVID-19 disease severity.

A full list of author affiliations appears at the end of the paper.

Autoantibodies have been identified in patients with coronavirus disease 2019 (COVID-19), suggesting that the infection by severe acute respiratory syndrome virus 2 (SARS-CoV-2) can display features similar to a systemic autoimmune disease[1–5]. For instance, high levels of antiphospholipid autoantibodies have been linked to severe respiratory disease by inducing neutrophil extracellular traps (NET) and venous thrombosis[4,6–9]. Further, high titers of neutralizing immunoglobulin G (IgG) autoantibodies against type I interferons (IFN) have been reported in patients with life-threatening COVID-19[10]. Most recently, a wide range of autoantibodies in patients with COVID-19 have been characterized using rapid extracellular antigen profiling (REAP)[11]. This is a technology that allows the comprehensive and high-throughput identification of autoantibodies by recognizing 2770 extracellular and secreted protein components of the exoproteome (extracellular protein epitopes)[12]. Wang et al.[11] showed that COVID-19 patients have multiple autoantibodies against the exoproteome. While patients with mild disease or asymptomatic infection exhibit increased autoantibody reactivity relative to uninfected individuals, those with severe disease have the highest reactivity scores.

These results are in line with our previous report[13] on autoantibodies targeting the largest superfamily of integral membrane proteins in humans[14], i.e., G protein-coupled receptors (GPCR), suggesting that these autoantibodies are natural components of human biology that become dysregulated in autoimmune diseases[15]. Our prior work indicated that GPCR-specific autoantibody signatures are associated with physiological and pathological immune homeostasis[13]. Likewise, recent studies have detected functional antibodies against GPCRs in the sera of patients with COVID-19 and have indicated that they may be associated with disease severity[16–18]. However, these investigations focused only on a few anti-GPCR autoantibodies. Importantly, they did not investigate their relationship with the potential presence of autoantibodies targeting other GPCRs and renin-angiotensin system (RAS)-related molecules, which play a central role in the development of severe COVID-19. Thus, we employ a systems immunology approach (Fig. 1a) to characterize the relationship between autoantibodies targeting a broad group of GPCRs and RAS-related molecules with COVID-19 severity by determining their correlation signatures across SARS-CoV-2-infected patients versus healthy individuals.

## Results

### Autoantibodies against GPCRs and the renin-angiotensin system (RAS)-related molecules.
Here, we investigated the serum levels of autoantibodies targeting molecules belonging to the RAS (including the GPCRs: MAS1, AGTR1, and AGTR2 as well as the entry receptor for SARS-CoV-2, angiotensin-converting enzyme II [ACE2])[19–22]. Furthermore, we assessed the concentrations of autoantibodies against GPCRs involved in chemotaxis and inflammation (CXCR3[23,24] and C5AR1[25]), coagulation (F2R[26]), and neuronal receptors (ADRA1A, ADRB1, and ADRB2, CHRMs)[27–31], which have been implicated in the development of COVID-19 disease (see Supplementary Data 1–3 for autoantibody levels as well as abbreviations of autoantibodies and their targets). In addition, we investigated autoantibodies targeting receptors facilitating the infectivity of SARS-CoV-2, and its entrance into host cells (neuropilin [NRP1]-aab)[32]. Finally, we explored the potential presence of autoantibodies against STAB1 (STAB1-aab), a scavenger receptor, as a potential new candidate in COVID-19 pathology since, despite the lack of investigations into its role in COVID-19, its multifunctionality during leukocyte trafficking, tissue homeostasis, and resolution of inflammation suggests that it could be relevant for disease severity[33,34].

Figure 1b and c display the interactions of these autoantibody targets represented by their physical protein-protein interaction (PPI) (Supplementary Data 4) and gene ontology (GO) relationships (Supplementary Data 5), respectively.

We found significantly higher levels of autoantibodies mostly directed against 11 receptors (AGTR1, AGTR2, ADRB1, BDKRB1, MAS1, CXCR3, CHRM3, CHRM5, NRP1, F2R, STAB1) in the moderate or severe COVID-19 groups than in the healthy control and mild COVID-19 group (Fig. 2a and b). These findings indicate that these autoantibodies reached the highest serum level in patients with moderate and severe disease (Fig. 2c). These receptors are involved in the modulation of inflammation and the RAS, suggesting that there could be relevant biological pathways which underly the identified associations between these autoantibodies and COVID-19 severity. In line with this, both controls and the COVID-19 disease groups (mostly mild COVID-19 patients) were found to have some autoantibodies at similar levels to lesser biologically relevant targets with respect to disease severity. Most of these autoantibodies were targeting neuronal receptors (e.g., ADRA1R, ADRB1, ADRB2, CHRM3, and CHRM4)[35–37], but also the receptor for complement C5a (C5AR1), a potent anaphylatoxin chemotactic receptor[38]. Thus, our data suggest that severe COVID-19 is associated with autoantibodies toward certain groups of GPCRs. Additionally, we found that the dysregulated production of autoantibodies targeting GPCRs and RAS-related molecules in COVID-19 patients was accompanied by higher levels of some autoantibodies associated with classic autoimmune diseases[39] when compared to healthy controls. For instance, while we found no significant differences in antinuclear antibodies (ANA), the levels of rheumatoid factor (RF), and autoantibodies targeting double-stranded DNA (anti-dsDNA) significantly increased according to COVID-19 severity (Fig. 2d). Of note, we did not include asymptomatic individuals infected with SARS-CoV-2 in this article because their sera were not available at the time of data acquisition for this study. However, we are currently performing a follow up study with a German cohort of COVID-19 patients and have so far observed that healthy controls fully recovered from COVID-19 have a pattern of autoantibodies targeting GPCRs and RAS-related molecules that resembles that from healthy controls and mild COVID-19 patients (manuscript in preparation).

### Autoantibody stratification of COVID-19 severity using multivariate analyses.
Next, we performed principal component analysis (PCA) using a spectral decomposition approach[40,41], to examine the correlations between variables (autoantibodies) and observations (individuals) while stratifying groups based on the autoantibody levels. This approach indicated that autoantibodies could be used to stratify COVID-19 patients according to disease severity (mild, moderate, and severe) (Fig. 3a–d). While healthy controls and patients with mild COVID-19 presented with more similar autoantibody patterns, moderate and severe COVID-19 patients clustered together. In this context, autoantibodies such as ACE2-aab, AGTR2-aab, BDKRB1-aab, CXCR3-aab, MAS1-aab, CHRM5-aab, NRP1-aab, F2R-aab, STAB1-aab appeared to play a major role in stratifying COVID-19 by disease burden. Altogether, these results indicate that the association between autoantibodies against GPCRs and COVID-19-related molecules can be used as biomarkers for COVID-19 outcomes.

### Machine learning classification of COVID-19 patients based on autoantibodies.
To further explore the potential of autoantibodies as biomarkers of COVID-19, we used random forest classification[42] based on autoantibody for predicting disease outcomes. The receiver operating characteristic (ROC) curve

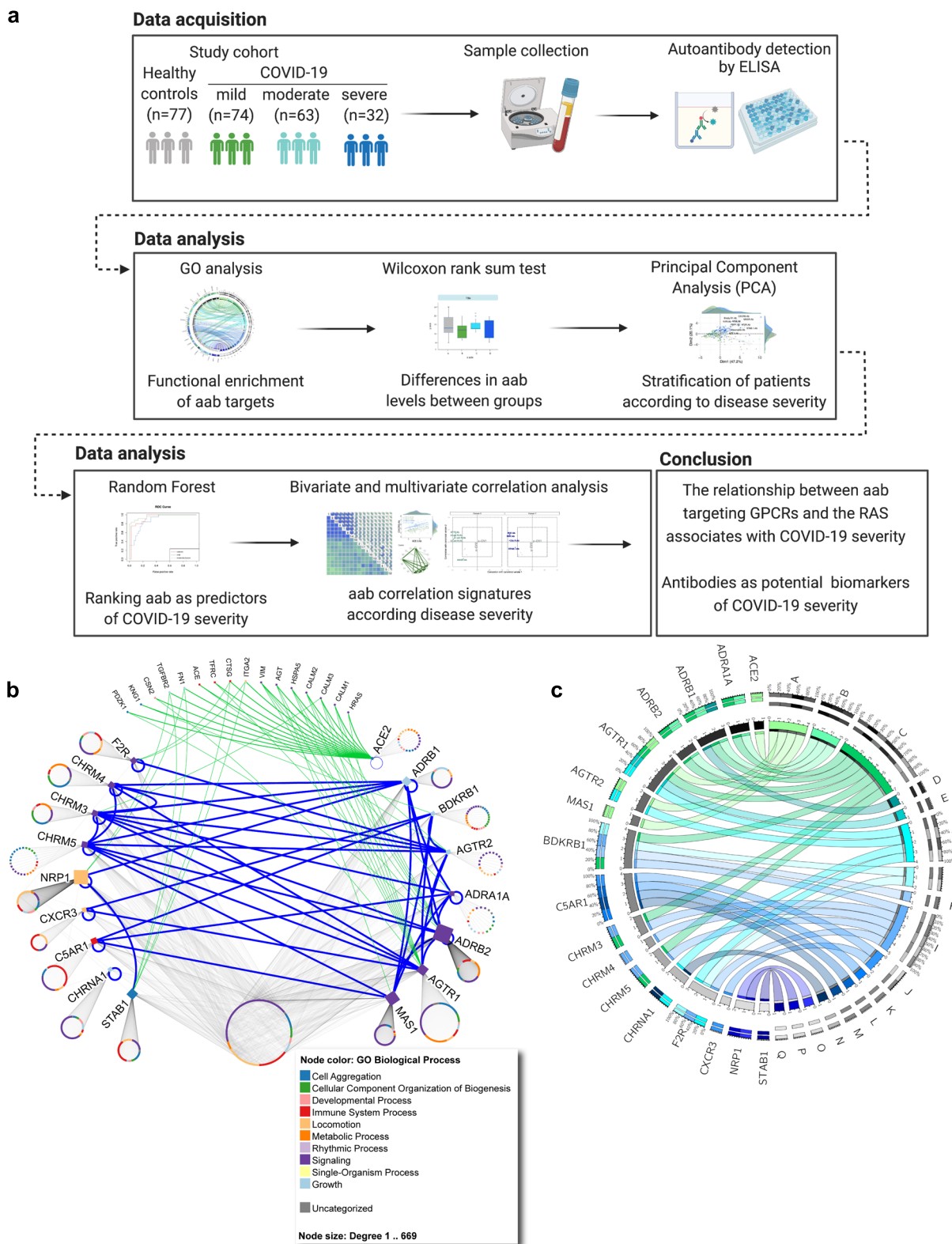

indicated a high false-positive rate for the classification of severe patients with the stable curve showing the highest error rate (out-of-bag or OOB) for this group (Fig. 4a and b). In accordance with the PCA, random forest classification of COVID-19 groups showed a higher error rate (low accuracy) when distinguishing moderate patients from those with severe COVID-19, suggesting that moderate and severe COVID-19 patients present a similar autoantibody pattern.

Thus, we assigned moderate and severe COVID-19 patients to the same group to identify the most relevant autoantibody predictors of COVID-19 burden. Using this approach, the merged moderate/severe patient group showed a lower error rate than the healthy controls and mild COVID-19 patients, reinforcing the previous observation that moderate and severe groups have an overlapping autoantibody pattern. This model resulted in an OOB error rate of 22.95% for all groups and areas

**Fig. 1 Study workflow. a** After data acquisition, we carried out different statistical analyses (written on the top) to characterize the signature of autoantibodies against GPCRs and COVID-19-associated molecules (e.g., renin-angiotensin system) in COVID-19 patients when compared with healthy controls. Created with BioRender.com. **b** Interaction network of autoantibody targets: molecules belonging or influencing the RAS (on the right) as well as additional molecules (other GPCRs, NRP1, and STAB1; on the left). The network highlights interactions among the autoantibody targets (blue edges), ACE-2 interactors connecting to the other targets (green edges), and gene ontology (GO) biological processes (node color). The number of interacting partners for each target is proportional to the node size. The circles associated with each autoantibody target are formed by their interactors, whose names are omitted. **c** Circos plot illustrating the functional relationships between the antibody targets and biological processes as indicated by GO enriched processes, which are denoted by letters: A renin-angiotensin system, B adrenergic signaling in cardiomyocytes, C calcium signaling, D renin secretion, E GP130/JAK/ STAT, F toll-like receptor signaling network, G complement and coagulation cascades, H inflammatory mediator regulation of TRP channels, I regulation of actin cytoskeleton, J inflammation mediated by chemokine and cytokine signaling, K immune system, L innate immune system, M neutrophil degranulation, N actions of nitric oxide in the heart, O human T-cell leukemia virus 1 infection, P VEGF and VEGFR signaling network, Q scavenging by class H receptors. The Circos plot shows only a few GO enriched processes; the complete list of relationships is provided in Supplementary Data 5. The size of the rectangles in the outer circles is proportional to the involvement of autoantibody targets in multiple pathways. The size of rectangles forming the inner circle represents genes and datasets with more connections to each other. Colors, numbers and percentages on the outer circles denote pleiotropy and gene-pathway associations. GO, gene ontology. Source data are provided as a Source Data file.

under the ROC curve of 0.93, 0.87, and 0.96 for the healthy control, mild COVID-19, and moderate/severe COVID-19 groups, respectively (Fig. 4c and d). Moreover, the random forest model ranked these 17 autoantibodies based on their ability to discriminate between healthy controls and COVID-19 disease severity groups. Follow-up analysis indicated that CXCR3-aab, AGTR1-aab, MAS1-aab, CHRM5-aab, and BDKRB1-aab were the five most significant predictors of COVID-19 disease-severity classification based on the number of nodes and the Gini decrease criteria for measuring variable importance (Fig. 4e and f). However, other autoantibodies such as F2R-aab and STAB1-aab were also strong predictors of COVID-19 severity. The interaction between anti-CXCR3 and anti-AGTR1 autoantibodies was the most frequent interaction occurring in the decision trees obtained by the random forest model (Supplementary Fig. 1). Altogether, these results show that autoantibodies targeting GPCRs and COVID-19-associated molecules perform well as predictors of COVID-19 disease severity.

Of note, the aforementioned results were adjusted for age and sex in the production of autoantibodies by randomly selecting age- and sex-matched healthy controls and COVID-19 patients, reducing the likelihood of confounding effects. As a further precaution, we also assessed whether sex and age were associated with the top 10 autoantibodies ranked as predictors of disease severity by random forest analysis (Supplementary Fig. 2A and B). Overall, except for the MAS1-aab, which was significantly higher in control females versus control males, there were no sex differences in the COVID-19 disease groups. We also further analyzed whether the use of medications was associated with the levels of these autoantibodies and observed significant changes in the levels of some autoantibodies in severe patients receiving vitamin C and zinc (Supplementary Fig. 3A–D). However, this observation requires future investigation, because the influence of several other variables could not be controlled for in our study such as the inclusion of placebo as well as time and dose–response groups. In this context, it will also be important to assess the relationship between autoantibody levels and peripheral lymphocyte counts to evaluate, for instance, the impact of changes in the number of circulating B lymphocytes on the serum levels of autoantibodies. Detailed information on the demographics, therapeutics and clinical outcomes of the cohort of SARS-CoV-2-infected patients is provided in Supplementary Data 1.

**Disruption of autoantibody correlation signatures in severe forms of COVID-19.** We recently reported that the hierarchical clustering signatures of anti-GPCR autoantibody correlations are associated with physiological and pathological immune homeostasis[13]. Based on this concept, we investigated the correlation signatures in healthy controls and patients with COVID-19 to explore whether changes in autoantibody relationships correlate with disease burden. Bivariate correlation analysis revealed a progressive loss of normal correlation signatures from mild to severe COVID-19 patients. In other words, patients with mild COVID-19 exhibited only minimal differences in the autoantibody correlation signatures when compared to healthy controls (Fig. 5a). Patients with moderate COVID-19 started to exhibit new relationships among autoantibodies while the severe group displayed the most different topological correlation pattern. Topologically, a positive correlation predominated among the autoantibodies. Of note, autoantibodies targeting nine different molecules presented with significant changes in the total correlation distribution, which was determined by the distribution of the pairwise correlation between autoantibodies (Fig. 5b and Supplementary Fig. 4). In summary, while the autoantibody network signatures were relatively conserved in patients with mild COVID-19 with respect to healthy controls, they were disrupted in moderate and most perturbed in patients with severe disease.

To better understand these changes in autoantibody correlation signatures, we performed canonical-correlation analysis (CCA), a multivariate statistical method that determines the linear relationship between two groups of variables[43]. CCA was carried out by splitting autoantibodies into two groups (as performed in Fig. 1b as well as Fig. 2a and b): those against molecules belonging or influencing the RAS (Dataset X) and those targeting other GPCRs, NRP1, and STAB1 (Dataset Y). This approach confirmed the changes in the autoantibody relationship patterns revealed by the bivariate correlation analysis. In addition, the CCA indicated changes based on COVID-19 severity were in agreement with the findings of the random forest model. For instance, in this multivariate correlation approach, autoantibodies targeting CXCR3 showed Spearman's rank correlation coefficients > 0.6 only in the moderate and severe groups (Fig. 5c). In this context, while BDKRB1-aab appeared only in the severe group, AGTR1-aab, MAS1-aab, and CHRM5-aab exhibited changes in their correlation patterns that were only observed in the severe group.

**Discussion**

Our work reinforces the idea that SARS-CoV-2 infection may trigger a life-threatening autoimmune disease, suggesting that this occurs against multiple molecules with key functions in immune and vascular homeostasis[1–3,44] such as GPCRs and RAS-related molecules. The precise mechanisms by which SARS-CoV-2 infection triggers the production of autoantibodies remain unknown. However, potential antigenic cross-reactivity (molecular

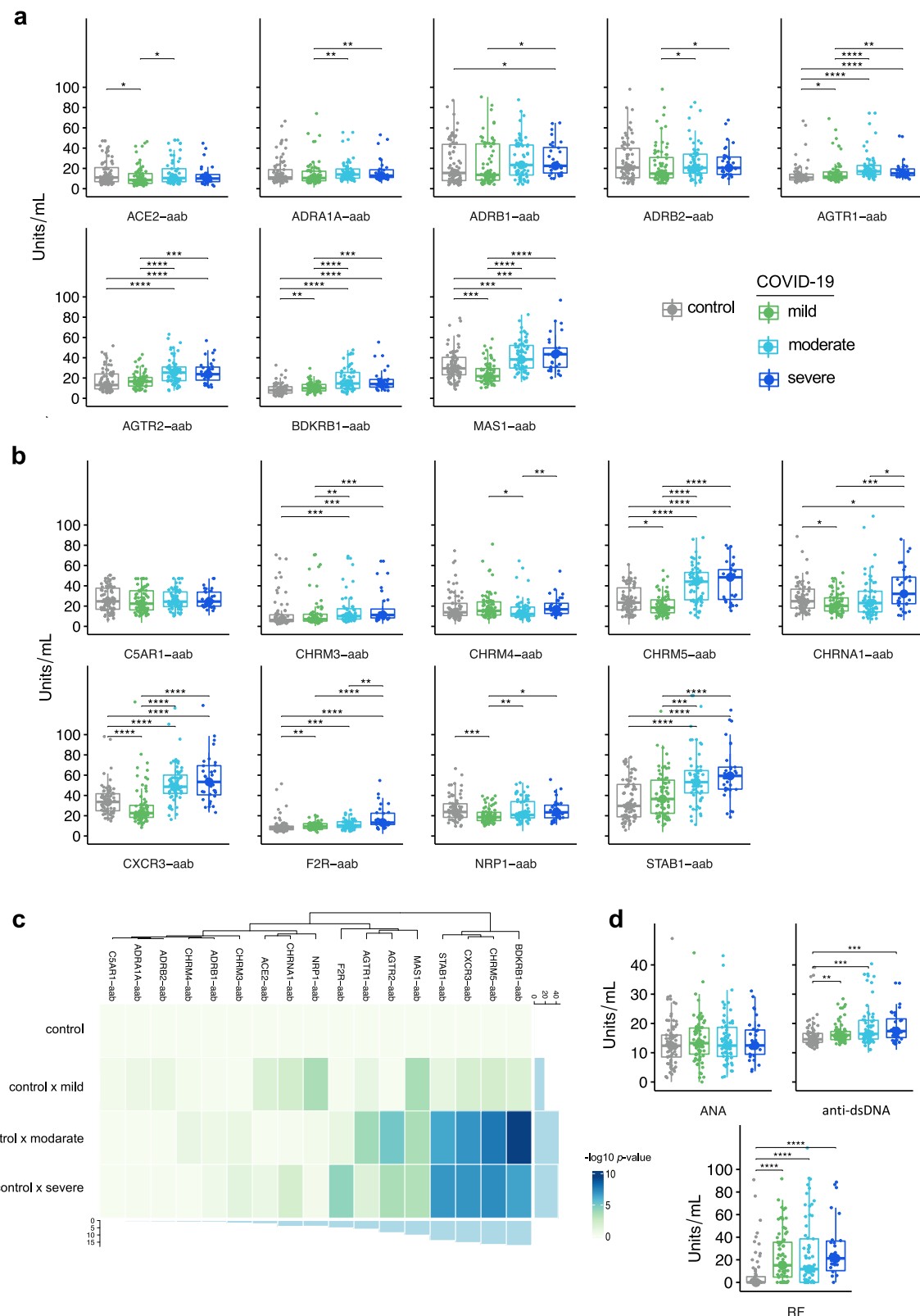

mimicry) between SARS-CoV-2 and human tissues has been hypothesized[45–50]. Furthermore, the hyperinflammatory reaction triggered by the virus results in tissue damage, causing systemic autoimmune-related manifestations that have been reported in patients with COVID-19[51]. In this context, while the mechanistic action of several autoantibodies that we identified remains to be investigated, we previously described[13,15,52,53] that anti-AGTR1[54] and anti-CXCR3 (previous work[55] and unpublished data) have agonist properties (e.g., on cell migration) and associate, for instance, with pulmonary fibrosis and cardiac death. Thus, these autoantibodies possibly potentialize the signaling triggered by their natural ligand, promoting the migration of immune cells, such as CD4+ and CD8+ T cells that are critical for both the killing of SARS-CoV-2 in the lung and deleterious hyperinflammation[56,57].

**Fig. 2 Autoantibodies against GPCRs and COVID-19-associated molecules are dysregulated during SARS-CoV-2 infection. a** and **b** Box plots of autoantibodies investigated in mild ($n = 74$), moderate ($n = 63$), and severe ($n = 32$) COVID-19 patients compared to healthy controls ($n = 77$): **a** autoantibodies against molecules belonging to or influencing the RAS; **b** autoantibodies targeting GPCRs and other molecules (NRP1-aab, and STAB1-aab). **c** Heatmap of −log10 p-value obtained from the comparisons of each COVID-19 group in relation to the control group. The bars aside the heatmap represent the sum of −log10 p-value. **d** Box plots of classical autoantibodies (antinuclear antibodies or ANAs; double-stranded DNA or dsDNA; and rheumatoid factor or RF) associated with autoimmune diseases. Each box plot shows the median with first and third interquartile range (IQR), whiskers representing minimum and maximum values within IQR, and individual data points. Significance was determined using two-sided Wilcoxon rank-sum tests and is indicated by asterisks (*$p \leq 0.05$, **$p \leq 0.01$, ***$p \leq 0.001$, and ****$p \leq 0.0001$). Source data are provided as a Source Data file.

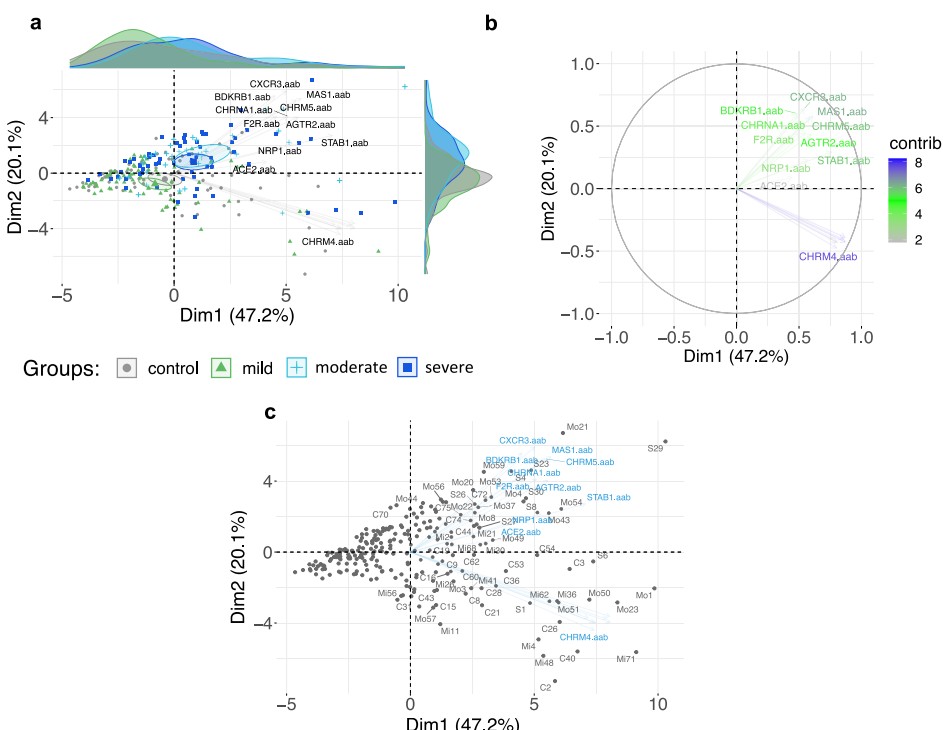

**Fig. 3 Autoantibodies stratify COVID-19 patients by disease severity. a** Principal component analysis (PCA) with spectral decomposition based on 17 different anti-GPCR-autoantibodies show the stratification of moderate ($n = 63$) and severe ($n = 32$) COVID-19 patients from mild ($n = 74$) COVID-19 patients and healthy controls ($n = 77$). Variables with positive correlation point to the same side of the plot, contrasting with negatively correlated variables, which point to opposite sides. Only autoantibodies highly contributing to the stratification of moderate and severe COVID-19 patients from mild patients and healthy controls are shown. Small circles are concentration ellipses around the mean points of each group. Histograms aside the PCA represent the density of the sample (individual) distribution. **b** Graphs of variables (antibodies) obtained by PCA of COVID-19 mild, moderate and severe groups and healthy controls, indicating the autoantibodies highly associated with moderate and severe COVID-19. The color scale bar indicates the contribution of each autoantibody to the principal component (PC). **c** Biplot of individuals (dark gray dots: c control, Mo moderate; Mi mild; S severe) and variables (autoantibodies: blue names) of same groups as in (**a**). Individuals with a similar autoantibody profile are grouped together. Healthy controls $n = 77$; COVID-19 groups: mild $n = 74$, moderate $n = 63$, and severe $n = 32$. Source data are provided as a Source Data file.

Regardless that we did not investigate the activity of autoantibodies on their targets, the results of our work underscore those of recent studies[3,6,10–12] that have reported the generation of autoantibodies following SARS-CoV-2 infection. Importantly, our data indicate that an additional immunopathological layer is present in which autoantibodies targeting GPCRs and RAS-related molecules are associated with COVID-19 burden. This association potentially sheds new light on the proposed immunopathological mechanisms underlying the development of COVID-19 infection, which is based on the abnormal activation of the ACE-II/angiotensin II (Ang II)/AGTR1/RAS axis together with a reduction of the ACE-II/angiotensin-(1-7)/MAS1 branch occurring together with several immunological dysregulation events[58].

The random forest model revealed an overlap between the autoantibody patterns of the moderate and severe COVID-19 groups, suggesting that an increase in autoantibody levels

accompanies progression from mild disease. Our cross-sectional study cannot show whether these antibodies were generated de novo. However, Chang et al.[59] reported a subset of antibodies against autoantigens similar to those observed in classic autoimmune diseases, as well as anti-cytokine antibodies that are generated de novo following SARS-CoV-2 infection. Accordingly, we have also identified higher levels of autoantibodies (anti-dsDNA and RF) in our COVID-19 cohort versus healthy controls and the details about their relationship with the clinical features of COVID-19 will be published elsewhere. Chang et al.[59] also showed that while some autoantibodies were at or below the average levels of healthy controls and increased over time during the SARS-CoV-2 infection, other autoantibodies were already elevated at the first time point of measurement in some seropositive patients, which is in accordance with the recently reported studies on preexisting autoantibodies to type I IFNs in

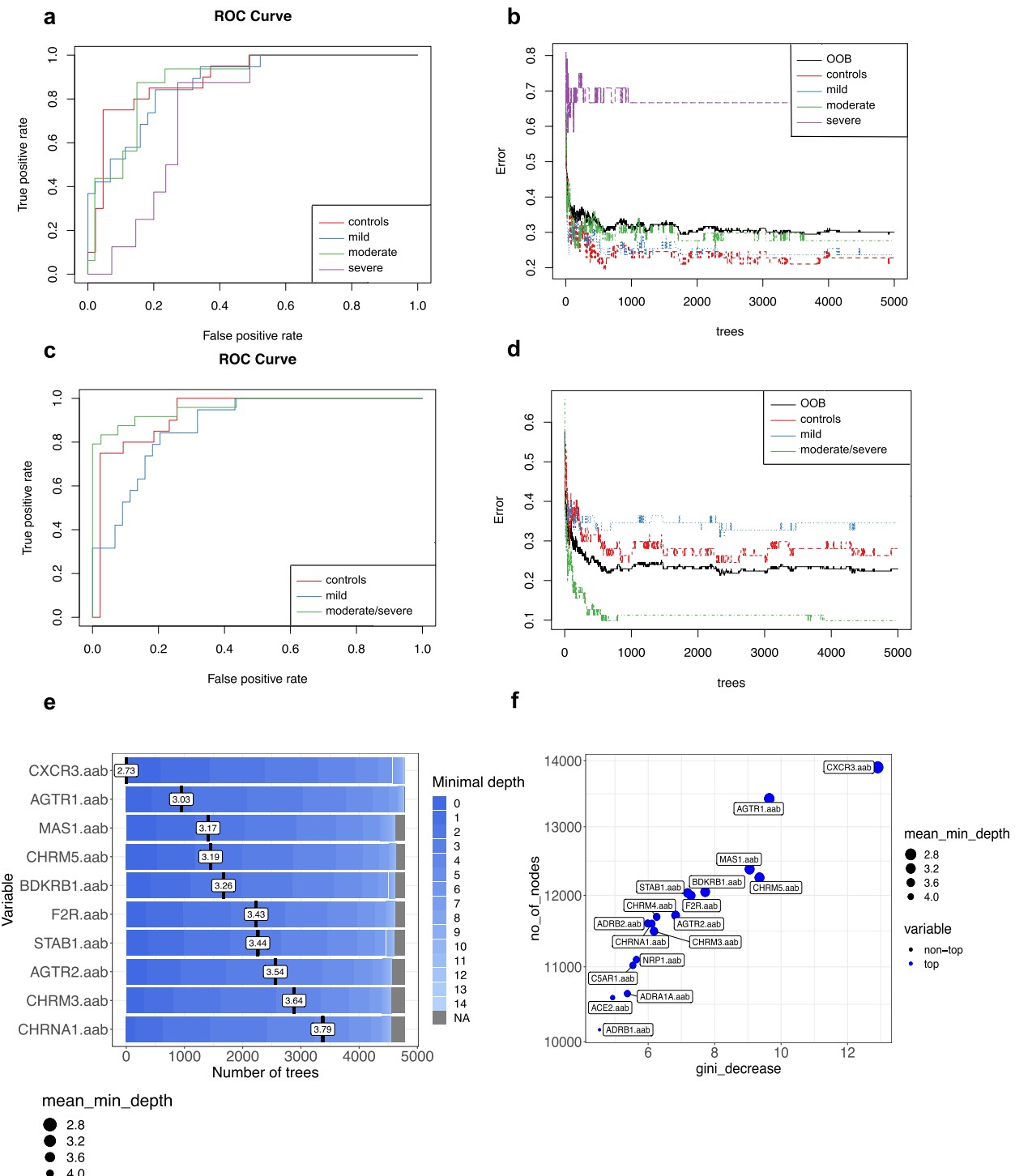

**Fig. 4 Ranking autoantibodies as predictors of disease severity reveals an overlap between their patterns in moderate and severe COVID-19.**
**a** Receiver operating characteristic (ROC) curves of 17 antibodies from mild ($n = 74$), moderate ($n = 63$), and severe ($n = 32$) COVID-19 patients versus healthy controls ($n = 77$) with an area under the curve (AUC) of 89.8% (for controls), 87.6% (for mild), 88,7% (for moderate), and 75.5% (for severe). **b** Stable curve showing number of trees and out-of-bag (OOB) error rate of 30.05%. **c** ROC curve of the same antibodies as in (**a**) from mild COVID-19 and moderate/severe COVID-19 patients compared to healthy controls with an AUC of 93.1% (for controls), 87.7% (for mild) and 96.2% (for moderate/severe). **d** Stable curve showing number of trees and OOB error rate of 22,95%. **e** Ranking of the top 10 autoantibody predictors of disease severity according to the mean minimal depth (black vertical bar with the mean value) calculated based on the number of trees. The blue color gradient reveals the minimum and maximum minimal depths for each variable. **f** Variable importance score plot based on Gini decrease and number (no.) of nodes for each variable showing which variable (antibody) presents a higher score in predicting COVID-19 severity. Source data are provided as a Source Data file.

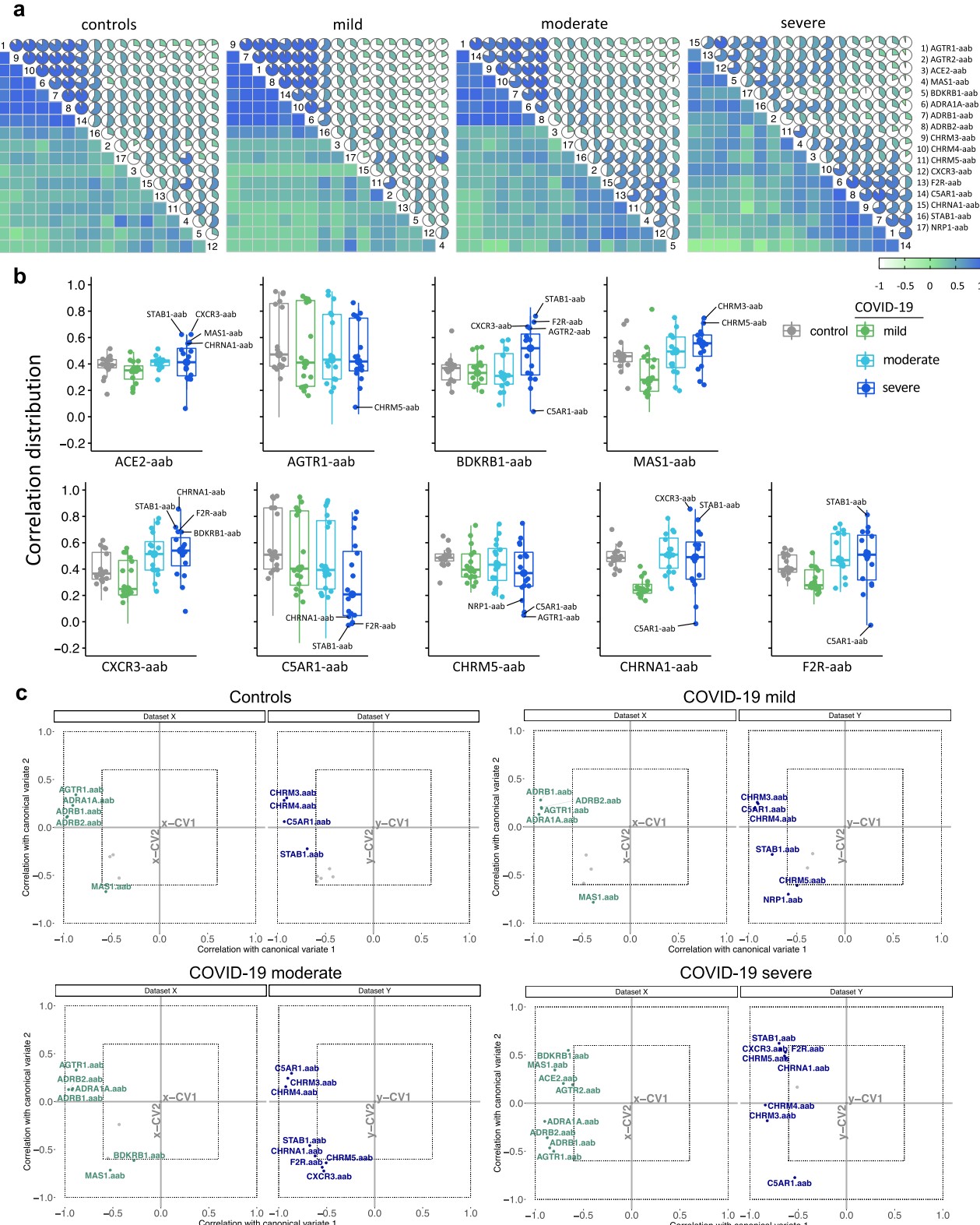

COVID-19 patients[60,61]. These data suggest that SARS-CoV-2 infection increases the production of autoantibodies or that at least some COVID-19 patients may have an unreported or undiagnosed pre-existing autoimmune disease. To the best of our knowledge, none of our patients had previously experienced autoimmune diseases and we do not have follow-up information available concerning the development of post-COVID autoimmune phenomena in the enrolled cohort. In this context, a previous report[13] of our group indicates that anti-GPCR autoantibodies are natural components of human biology that can dysregulate and trigger the development of autoimmune diseases (a concept discussed in detail elsewhere)[15,62]. Thus, we cannot exclude the possibility that at least some of our patients had dysregulated levels of autoantibodies targeting GPCRs and RAS-related molecules prior to SARS-CoV-2 infection. Thus, a limitation of our report that needs further investigation is the lack of

**Fig. 5 Autoantibody correlation signatures associate with disease burden. a** Correlation matrices of autoantibodies targeting GPCRs and the RAS (denoted by numbers as per legend) for the control ($n = 77$) and COVID-19 groups (mild [$n = 74$], moderate [$n = 63$], and severe [$n = 32$]). The color scale bar represents the range of Spearman's rank correlation coefficient. **b** Box plots illustrating the correlation distribution of autoantibodies with significant changes (as defined in Supplementary Fig. 4) in pairwise correlations: those belonging to the RAS are placed in the upper row, and autoantibodies targeting other GPCRs are exhibited in the lower row. Antibodies with the highest or lowest correlations and thus contributing more to changes in the correlation pattern of the severe COVID-19 group are indicated. Each box plot shows the median with first and third interquartile range (IQR), whiskers representing minimum and maximum values within IQR, and individual data points. **c** Canonical-correlation analysis (CCA) of autoantibodies. Correlation between autoantibodies against molecules belonging to or influencing the RAS (dataset X, in green) versus the other autoantibodies (those targeting other GPCRs, NRP1, and STAB1; dataset Y, in blue). Only autoantibodies with Spearman's rank correlation coefficient ≥ 0.6 are shown while those with a correlation coefficient < 0.6 (gray points) have their names omitted. Autoantibody correlations are plotted based on their relation to the first 2 canonical variates (x-CV1 and x-CV2; y-CV1 and y-CV2: ranging from −1 to 1). Autoantibodies located close in the same CCA quadrant region are those with the highest Spearman's rank correlation coefficient. Source data are provided as a Source Data file.

a longitudinal analysis of anti-GPCR/-RAS antibodies to evaluate their levels from disease onset until convalescence. Further, any potential link to post-acute COVID-19 syndrome remains to be investigated.

Our results suggest that autoantibodies targeting CXCR3 and AGTR1 are the most important predictors of COVID-19 severity. There is an essential biological connection between CXCR3 and AGTR1; blocking AGTR1 impairs the release of several chemokines, including CXCL10, the ligand for CXCR3[63], a chemokine receptor highly expressed by effector T cells controlling the trafficking and function of CD4+ and CD8+ T cells during inflammation[64–67]. Furthermore, both CXCR3 and AGTR1 have been strongly associated with both autoimmune and inflammatory diseases[68,69]. Additionally, increased levels of angiotensin II together with the hyperactivation of its receptor (AGTR1) have been associated with unfavorable COVID-19 disease[17,70]. This pathological mechanism has been explored as a therapeutic approach for COVID-19 by clinical trials with losartan, an AGTR1 antagonist[8,71]. AGTR1 orchestrates several important immunological functions and losartan treatment has been previously demonstrated to have immunomodulatory properties. Angiotensin II is the main effector molecule of the RAS that, upon binding to AGTR1, promotes vasoconstriction, inflammation, oxidative stress, coagulation, and fibrosis, all of which play an important pathological role during SARS-CoV-2 infection[20].

We also found an alteration in the normal relationship between autoantibodies targeting GPCRs and RAS-related molecules that was associated with COVID-19 severity by increasing disruption of autoantibody correlations according to disease burden. This observation provides new insights into the biology of autoantibodies, which is in line with our previous observation that GPCR-specific autoantibody signatures are associated with physiological and pathological immune homeostasis[13]. Since GPCRs comprise the largest superfamily of integral membrane proteins in humans[14], it is also possible that several additional anti-GPCR autoantibodies remain to be discovered. Likewise, several SARS-CoV-2 strains have been identified[72] and it will be important to investigate whether they induce different autoantibody patterns that may contribute to disease outcome. Of note, autoantibodies are present in healthy individuals and immunization with GPCR-overexpressing membranes can induce the production of autoantibodies targeting GPCRs[13]. Thus, another important issue to be addressed is whether the recently developed vaccines against COVID-19[73] could influence the production of anti-GPCR autoantibodies.

Overall, although we postulate that dysregulated autoantibodies targeting GPCRs and RAS-related molecules represent a pathological autoimmune phenomenon, it is also possible that some of them may have neutralizing activities, which requires future investigation. Considering the role of the immune system in homeostasis beyond host defense[74–76], these autoantibodies

could also represent both a physiological attempt of the immune system to promote body homeostasis during SARS-CoV-2 infection. In conclusion, this study identifies new autoantibodies that are dysregulated by SARS-CoV-2. Our data also indicate that anti-GPCR antibodies represent potential new clinically relevant biomarkers that predict COVID-19 severity. The increasing disruption of autoantibody network signatures moving from patients with mild, to moderate and finally severe disease suggests a gradual loss of autoantibody homeostasis that accompanies the progression of COVID-19 triggered by the SARS-CoV-2-induced immune dysregulation. Since a better understanding of the COVID-19 pathogenesis may open new avenues to improve diagnostic and therapeutic options[77,78], the results reported here may provide new insights to improve the clinical management of COVID-19 patients.

## Methods

**Patient cohort**. We included 246 adults from Jewish communities across 5 states of the United States of America. Among them, there were heathy controls and patients who had developed symptomatic COVID-19 disease before receiving any SARS-CoV-2 vaccine. The patients participated in an online survey developed to determine the most common symptoms and outcomes of SARS-CoV-2 infection[79,80]. Details about the survey study, patient demographics and symptoms have been previously described[79,80] and are present in Supplementary Data 1. Seventy-seven randomly selected age- and sex-matched healthy controls (SARS-CoV-2 negative and without COVID-19 symptoms) were included in this study and their autoantibody data were compared to those of 169 individuals who were SARS-CoV-2 positive (determined by positive nasopharyngeal swabs). The SARS-CoV-2 infected cohort were divided into mild COVID-19 ($n = 74$; fever duration ≤ 1 day; peak fever of 37.8 °C), moderate COVID-19 ($n = 63$; fever duration ≥ 7 day; peak fever of ≥38.8 °C) and severe COVID-19 groups ($n = 32$; severe symptoms and requiring supplemental oxygen therapy). Disease severity for SARS-CoV-2-positive individuals was determined based on the World Health Organization (WHO) severity classification[81]. All healthy controls and all patients provided written consent to participate in the study, which was performed in accordance with the Declaration of Helsinki and approved by the IntegReview institutional review board. In addition, this study followed the Strengthening the Reporting of Observational Studies in Epidemiology (STROBE) reporting guideline.

**Detection of IgG autoantibodies**. Human IgG autoantibodies against 14 different GPCRs (AGTR1, AGTR2, MAS1, BDKRB1, ADRA1A, ADRB1, ADRB2, CHRM3, CHRM4, CHRM5, CXCR3, F2R, C5AR1, CHRNA1), 2 molecules serving as entry for SARS-CoV-2 (ACE2, NRP1), and antibodies against the transmembrane receptor STAB1 were detected from frozen serum using commercial ELISA kits (CellTrend, Germany) according to the manufacturer's instructions (https://www.celltrend.de/), as previously described[55]. Briefly, duplicate samples of a 1:100 serum dilution were incubated at 4 °C for 2 h. The autoantibody concentrations were calculated as arbitrary units (U) by extrapolation from a standard curve of five standards ranging from 2.5 to 40 U/ml. The ELISA kits were validated according to the Food and Drug Administration's Guidance for Industry: Bioanalytical Method Validation. Autoantibodies associated with classic autoimmune diseases (ANA, RF, anti-dsDNA) were also measured using commercial ELISA kits according to the manufacturer's instructions (Inova Diagnostics, San Diego, CA, USA).

**Interaction network and enrichment analysis of autoantibody targets**. We used IID ver 2021-05[82] to search for physical protein interactions of the autoantibody

targets, which was used to build a network figure prepared using NAViGaTOR version 3.0.15[83]. NAViGaTOR was also used to visualize the interactions of autoantibody targets and highlight their GO biological processes (node color), direct interactions among the 17 autoantibody targets (blue edges), and ACE-2 interactors connecting to the other targets (green edges). The network was then exported in an SVG format, and the final TIFF image with legends was prepared in Adobe Illustrator ver 26.0. All protein interactions with annotation are available in Supplementary Data 2. Comprehensive pathway analysis of the 17 autoantibody targets and their interactors was performed using pathDIP ver 4.1[84]. A circos plot of antibody targets and pathway associations was built using the Circos online tool[85].

**Differences in autoantibody levels**. Box plots showing the different expression levels of 17 anti-GPCR-autoantibodies from COVID-19 patients (mild, moderate and severe groups) and healthy controls were generated using the R version 4.0.5 (The R Project for Statistical Computing. https://www.r-project.org/), R studio Version 1.4.1106 (R-Studio. https://www.rstudio.com/), and the R packages ggpubr, lemon, and ggplot2. Statistical differences in autoantibody levels were assessed using a two-sided Wilcoxon rank-sum test as previously described[11].

**Principal component analysis**. PCA with spectral decomposition[40,41] was used to measure the stratification power of the 17 autoantibodies in distinguishing between COVID-19 (mild, moderate, and severe patients) and healthy controls. PCA was performed using the R functions prcomp and princomp, through factoextra package (principal component analysis in R: prcomp vs. princomp)[86].

**Machine learning model and autoantibody ranking**. We employed random forest model to construct a classifier for discriminating among controls and mild, moderate, and severe COVID-19 patients. This approach aimed to identify the most significant predictors for severe COVID-19. We trained the random forest model using the functionalities of the R package randomForest (version 4.6.14)[87]. Five thousand trees were used, and three variables were resampled. Follow-up analysis was conducted with the Gini decrease, number of nodes, and mean minimum depth as criteria to determine variable importance. The adequacy of the random forest model as a classifier was assessed through the out-of-bags error rate and the ROC curve. For cross-validation, we split the dataset into training and testing sets, using 75% of the observations for training and 25% for testing.

**Autoantibody correlation signatures: bivariate and multivariate correlation analysis**. Bivariate correlation analysis of autoantibodies for each group (controls and mild, moderate, and severe COVID-19 patients) was performed using the corrgram, psych, and inlmisc R packages. In addition, multilinear regression analysis of the relationships between different variables (autoantibodies) was performed using the R packages ggpubr, ggplot2 and ggExtra. CCA[88] of autoantibodies against molecules associated with the RAS, other GPCRs and SARS-CoV-2 entry molecules was performed using the R packages CCA and whitening[88]. CCA is a classic statistical tool for performing multivariate correlation analysis. We used log-transformed antibody levels to perform both bivariate correlation and CCA analysis.

**Reporting summary**. Further information on research design is available in the Nature Research Reporting Summary linked to this article.

## Data availability

A reporting summary for this article is available as a Supplementary Information file. All data generated in this study are provided in the Supplementary Data/Source Data files. The source data underlying the Main and Supplementary Figures are provided as a Source Data file. Source data are provided with this paper.

## Code availability

All R packages used in this manuscript are described in the Reporting Summary and are available at the following link: https://github.com/lschimke/The-relationship-between-autoantibodies-targeting-GPCRs-and-the-renin-angiotensin-system-associates-

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

## Acknowledgements

We acknowledge the patients for participation in this study. We also thank the São Paulo Research Foundation (FAPESP grants: 2018/18886-9, 2020/01688-0, and 2020/07069-0 to O.C.M.; 2020/09146-1 to P.P.F.; 2020/16246-2 to D.L.M.F.; 2020/07972-1 to G.C.B., 2020/11710-2 to D.R.P.), and the Coordination for the Improvement of Higher Education Personnel (CAPES) Financial Code 001 (grant to ISF) for financial support. We acknowledge the Ontario Research Fund (grant #34876), Natural Sciences Research Council (NSERC #203475), Canada Foundation for Innovation (CFI #29272, #225404, #33536), and IBM. This work was also supported by the Deutsche Forschungsgemeinschaft (DFG) founding the Excellence Cluster Precision Medicine in Inflammation, project TI4 and CD1, by the COVID fund of Schleswig-Holstein as well as by DFG project RI 1056 11-1/2. This work was

supported by the Bundesministerium für Bildung und Forschung [01EC1901D (MESIN-FLAME)]. We thank Prof. Dr. med. Tanja Lange from the Department of Rheumatology and Clinical Immunology at the University of Lübeck, Germany for her advice to measure autoantibodies targeting the angiotensin-(1-7) receptor MAS1. We would like to acknowledge the contributions of Lev Rochel Bikur Cholim of Lakewood (led by Rabbi Yehuda Kasirer and Mrs. Leeba Prager) and their hundreds of volunteers who participated in collecting samples for this research.

## Author contributions

O.C.M., L.F.S., G.H., A.Z.R., G.R., Y.S. wrote the manuscript; O.C.M., L.F.S., A.Z.R., S.S., I.J., C.S., R.D., J.Y.H., A.M., L.M.G., G.H., H.G., A.S., F.T., Y.O., A.V., H.D.O., H.H., K.S.F., A.H., G.R., Y.S. provided scientific insights; O.C.M., L.F.S., I.J., Y.B.L., A.H.C.M., I.S.F., D.R.P., G.C.B., P.P.F., D.L.M.F. performed data and bioinformatics analyses; J.I.S., A.Z.R., I.Z., M.T.L. diagnosed, recruited or followed-up the patients; H.H., J.J., Y.O., A.V., C.M., A.M., K.S.F. coordinated the serum collection and databank or performed the experiments. H.H., H.A., J.Z., Y.O., A.V., A.Z.R., G.R. and Y.S. conceived the project and designed the study; O.C.M., G.H., L.F.S., I.J., C.S., F.T., Y.O., A.Z.R., L.M.G., Y.B.L., G.R., Y.S., H.D.O. revised and edited the manuscript; A.Z.R., G.R. and Y.S. supervised the project.

## Competing interests

The authors declare that H.H. and K.S.F. are CellTrend managing directors and that GR is an advisor of CellTrend and earned an honorarium for her advice between 2011 and 2015. The other authors declare no competing interests.

## Additional information

[1]Department of Immunology, Institute of Biomedical Sciences, University of São Paulo, São Paulo, SP, Brazil. [2]Department of Clinical and Toxicological Analyses, School of Pharmaceutical Sciences, University of São Paulo, São Paulo, SP, Brazil. [3]Network of Immunity in Infection, Malignancy, and Autoimmunity (NIIMA), Universal Scientific Education and Research Network (USERN), Sao Paulo, Brazil. [4]Zabludowicz Center for Autoimmune Diseases, Sheba Medical Center, Tel-Hashomer, Israel. [5]Saint Petersburg State University, Saint-Petersburg, Russia. [6]Ariel University, Ariel, Israel. [7]Department of Immunology, Immunosciences Laboratory, Inc., Los Angeles, CA, United States. [8]Cyrex Laboratories, LLC 2602S. 24th St., Phoenix, AZ 85034, USA. [9]Department of Pediatrics, NYU Langone Medical Center, New York, NY, USA. [10]Maimonides Medical Center, Brooklyn, NY, USA. [11]Department of Biology, Yeshiva University, Manhatten, NY, USA. [12]Department of Internal Medicine I, University Medical Center Schleswig-Holstein Campus Kiel, Kiel, Germany. [13]Department of Rheumatology, University Medical Center Schleswig-Holstein Campus Lübeck, Lübeck, Germany. [14]Department of Internal Medicine, Haraldsplass Deaconess Hospital, Bergen, Norway. [15]CellTrend Gesellschaft mit beschränkter Haftung (GmbH), Luckenwalde, Germany. [16]Department of Pediatrics, University of Washington School of Medicine, and Seattle Children's Research Institute, Seattle, WA, USA. [17]Sackler Faculty of Medicine, Tel-Aviv University, Tel-Aviv, Israel. [18]Institute of Medical Immunology, Charité - Universitätsmedizin Berlin, Corporate Member of Freie Universität Berlin, Humboldt-Universität zu Berlin, and Berlin Institute of Health, Berlin, Germany. [19]Experimental and Clinical Research Center, a collaboration of Max Delbruck Center for Molecular Medicine and Charité Universitätsmedizin, and HELIOS Clinic, Department of Cardiology and Nephrology, Berlin 13125, Germany. [20]Osteoarthritis Research Program, Division of Orthopedic Surgery, Schroeder Arthritis Institute, UHN; Data Science Discovery Centre, Krembil Research Institute, UHN, Departments of Medical Biophysics and Computer Science, University of Toronto, Toronto, Canada. [21]Institute of Neuroimmunology, Slovak Academy of Sciences, Bratislava, Slovakia. [22]School of Medicine and Health Sciences, George Washington University, Washington, DC, USA. [23]Department of Medicine B, Sheba Medical Center, Tel Hashomer, Israel. [24]Department of Pathology, Johns Hopkins University, Baltimore, MD, USA. [25]These authors contributed equally: Otavio Cabral-Marques, Gilad Halpert, Lena F. Schimke, Gabriela Riemekasten, Yehuda Shoenfeld. ✉email: otavio.cmarques@usp.br; gabriela.riemekasten@uksh.de; shoenfel@post.tau.ac.il

