## [Peer Review File · Nature Communications]

Autoantibodies targeting GPCRs and RAS-related molecules associate with COVID-19 severityREVIEWER COMMENTS

Reviewer #1 (Remarks to the Author):

Manuscript title: The relationship between autoantibodies targeting GPCRs and the renin-angiotensin system associates with COVID-19 severity

Comments/suggestions: I believe the following comments will help the authors to improve the scientific merit of this manuscript:

- Figure 1(A): All nodes have an equal size. For a better network presentation, node sizes should be assigned in proportion to their degree centrality values.
- Figure 2: PCA is okay. It seems there is a clustering technique applied in A (small circle), but not mentioned in the title. The meaning of colouring in C is not clear
- Figure 3: differences in grouping between B (moderate and severe separate group) and D (same group). Any reason? Need to check the finding from D with separate groupings (i.e., consider moderate and severe groups separately). Also, need to change the figure title and corresponding in-text description/discussion accordingly.
- Supple. Figure 2 (B): This could be a decision tree within the forest. The underlying process for completing the classification job for the RF is more complex. The authors either avoid such a presentation or make it more apparent in the figure title.
- Suppl. Figure 4: A 2D correlation plot would be used for the same purpose. A network presentation is not relevant here, especially from the 'Network analysis' viewpoint.

Reviewer #2 (Remarks to the Author):

The manuscript by Cabral-Marques et al. investigates the relationship between autoantibodies against GPCRs or RAS-related molecules and COVID-19 outcomes.

In this cross-sectional study 169 SARS-CoV-2-infected individuals and 77 matched uninfected controls were compared for their autoantibody profile. Results showed that disease severity is significantly associated with highest autoantibody levels and that antibodies reacting against downstream molecules in the RAS cascade as well as CXCR3 have the strongest association with severe outcome.

The manuscript is well-written and relevantly contributes to deepen the knowledge of COVID-19 pathogenesis.

I have some minor points:

-Due to the cross-sectional design of the study, it is unclear whether such autoantibodies were pre-existent or developed in response to SARS-CoV-2 infection. Were there any patients in the enrolled cohort who suffered from previous autoimmune diseases? And is any information available concerning the development of post-COVID autoimmune phenomena in the enrolled cohort? Did patients develop other "classical" autoantibodies during SARS-CoV-2 infection (e.g., ANA, aPL, ENA, RF, etc.?). That would be of crucial importance in connecting SARS-CoV-2 infection with the risk of full-blown autoimmune disorders in a more generalized virus-induced loss of self-tolerance.

-Although demographics, therapeutics and clinical outcomes of the whole original cohort of SARS-CoV-2-infected patients have been described elsewhere (ref. 77,78), it would be useful to readers to report such data for the 169 selected patients and 77 controls in a table. It seems that asymptomatic SARS-CoV-2-infected individuals were not included in the analysis. The authors should motivate the reason for this choice.

- Have gender, age, peripheral lymphocyte count (and relative lymphocyte subsets) and medications (e.g., steroids, antimalarials, immunosuppressive or biological agents) influenced the synthesis of these autoantibodies and to what extent?

- The discussion section provides several hypotheses for the pathogenicity of autoantibodies targeting the RAS and GPCRs withing other pro-inflammatory circuits in COVID-19, but no mention is made of neutralizing and binding antibodies that may indeed affect physiological pathways in a distinct way. Please clarify and update this information in the method section.

- In the main section, please update the results by Wang et al. saying that both asymptomatic SARS-CoV-2-infected individuals (rather than healthy controls) and COVID-19 patients have multiple antibodies against the exoproteome.

- In the discussion, the authors report that patients with moderate COVID-19 symptoms present with strong antibody production and high titers of neutralizing antibodies (ref. no. 50). This is only in part true. In their paper, Garcia-Beltran et al. rather write that "altogether, severity of SARS-CoV-2 infection significantly correlates with higher anti-RBD antibody levels, but suboptimal neutralization potency is a significant predictor of mortality".

- In the patient cohort section, please provide a reference for the WHO severity classification of COVID-19.

- Please note that figure 4 is erroneously miscalled as figure 5 in the legend.

- Beside controls, Suppl. Table 1 reports a list of COVID-19 patients with mild, severe and oxygen-dependent disease but no patients with moderate disease. Please verify.

Reviewer #3 (Remarks to the Author):

Key points:

This study by Cabral-Marques and colleagues draws on previous work from this group, notably the finding of autoantibodies targeting the G protein-coupled receptors (GPCRs), proposing that these autoantibodies are natural constituents of human biology that become dysregulated in autoimmune diseases. Given that Covid-19 infection often presents with immunologic features, these investigators sought to determine whether such autoantibodies were present, and what the clinical significance was based on their presence and extended the work to autoantibodies targeting the immune and the renin-angiotensin systems (RAS) including GPCRs: MASR, AT1R, and AT2R in addition to ACE-II, which play an essential role in the development of severe COVID-19. Furthermore, the concentration of autoantibodies targeting GPCRs involved in chemotaxis, inflammation, and coagulation were studied, as well as neuronal receptor antibodies and antibodies targeting neuropilin, and STAB1.

Drawing on their previous reported that hierarchical clustering signatures of anti-GPCR autoantibody correlations are associated with physiological and pathological immune homeostasis, they investigated the autoantibody "correlation signatures" in healthy controls and patients with COVID-19 to determine if changes in autoantibody relationships correlate with disease severity.

Insight:

The paper shows that autoantibody network signatures were relatively conserved in patients with mild COVID-19 compared to healthy controls, and such signatures were disordered in moderate and most perturbed in severe patients. Canonical-correlation analysis (CCA) was added to the Bivariate correlation analysis to try to better discern the autoantibody correlation signatures.

Significance

The precise mechanisms by which SARS-CoV-2 infection prompts the production of autoantibodies remains unknown. When compared with patients manifesting mild disease, those with moderate COVID-19 symptoms demonstrate increased production of autoantibodies. The authors contend that their data show autoantibodies targeting GPCRs and RAS-related molecules are associated with COVID-19 burden. They assert that the association provides further evidence for the the proposed immunohematological mechanisms underlying the development of COVID-19 infection (which is grounded in the proposed abnormal activation of the ACE-II/Angiotensin II (Ang II)/AT1R/RAS axis together with a decrease of the ACE-II/ Angiotensin-(1-7)/MASR branch.

They suggest that their data provides new insights into the biology of autoantibodies, which they contend is in line with their previous observation that "GPCR-specific autoantibody signatures associate with physiologic and pathologic immune homeostasis."

They postulate several possible mechanism including the targeting of new epitopes during severe Covid-19.

Future provocative studies include whether the production of anti-GPCR autoantibodies is .

These are potentially significant and provocative findings and assertions; however, they rely on the accuracy and conclusions of the statistical modeling provided. Additionally, it is unclear how such modeling can be clinically useful unless an online calculator is provided, or there is another way to translate a "risk score" based on autoantibody correlations.

Some mechanistic explanations for the study's findings are proposed but not yet proven.

Clarity and context:

The methods were appropriate, notably regarding patient selection and matching to controls. The text is clearly written and easily accessible with regard to the prose but the complicated mathematical modeling, while explained, is difficult to follow for non-statisticians or those with

minimal biostatistical background training. .

References

The references appear to be appropriate and correct although this paper is predicated largely upon this group's prior work, and there is less consensus outside of their findings that actually verifies their prior findings, That is a troublesome aspect.

Display items

The tables and figures are nicely displayed but difficult for a non-statistician to follow. I think the authors attempt to walk the unseasoned reader through their findings and potential significance but in the absence of a biostatistical background, the data is hard to follow.

Suggested improvements

I have limited suggestions for the authors as I think the manuscript reads well; however, it relies on the prior findings of the group as well as a biostatistical analysis that may be generally inaccessible to most readers. Reproducibility by other investigators would also lend more credibility to these findings.

Minor suggestions:

(1) This group has previously reported on autoantibodies targeting the G protein-coupled receptors (GPCRs) suggesting that these autoantibodies are natural components of human biology that become dysregulated in autoimmune diseases. This information should be incorporated into the abstract for context as to what these were ascertained in Covid-19 patients.

(2) Last paragraph under MAIN section: "However, these investigations were not systemic" – I think the authors mean to say systematic. If not, they need to clarify what they mean by "systemic" here.

(3) Third paragraph under DISCUSSION section: "Furthermore, our work indicates a change in the relationship between autoantibodies targeting GPCRs and RAS that associate with COVID-19 severity, which was shown by increasing disruption of autoantibody correlations according disease burden." The word "to" is missing before the word disease.

Expertise:

The statistical analyses here are outside my scope of expertise and I strongly suggest that a biostatistician verify the analyses and claims of the significance of the results.

Point-by-point response

Manuscript NCOMMS-21-33172: "The relationship between autoantibodies targeting GPCRs and the renin-angiotensin system associates with COVID-19 severity"

Thank you for the positive review of our manuscript and we appreciate the constructive criticism from the reviewers. Please, find below a detailed point-by-point response to the reviewers' comments.

Reviewer #1

Comments/suggestions: I believe the following comments will help the authors to improve the scientific merit of this manuscript:

We appreciate the comments/suggestions from Reviewer #1. We followed his/her comments and they really helped to improve our manuscript.

– Figure 1(A): All nodes have an equal size. For a better network presentation, node sizes should be assigned in proportion to their degree centrality values.

We replaced the original Figure 1A (now **Figure 1B**). The nodes are assigned in proportion to their centrality degree. In addition, we included the association of each autoantibody targets with their interactors and enriched gene ontology biological processes.

– Figure 2: PCA is okay. It seems there is a clustering technique applied in A (small circle), but not mentioned in the title. The meaning of colouring in C is not clear.

We added the information in the legend of this figure (now **Figure 3**): “Small circles are concentration ellipses around each group mean points. However, no cluster technique was applied in this approach.

– Figure 3: differences in grouping between B (moderate and severe separate group) and D (same group). Any reason? Need to check the finding from D with separate groupings (i.e., consider moderate and severe groups separately). Also, need to change the figure title and corresponding in-text description/discussion accordingly.

We performed the analysis grouping between B (moderate and severe separate group) and D (same group) because we suspected that moderate and severe groups have an overlap in their autoantibody patterns. We confirmed this possibility when carrying out the Random Forest analysis. We changed the figure title accordingly: “Ranking autoantibodies as predictors of disease severity reveals an overlap between their patterns in moderate and severe COVID-19”. We also included this information in the Result (**lines 108 to 117**) and Discussion (**lines 199-202**) sections. In addition, we further show this issue in a new Figure (**Figure 2C**), which exhibits the similarities between the autoantibody patterns observed in moderate and severe COVID-19 patients and their differences in relation to the healthy control and mild COVID-19 groups.

– Supp. Figure 2 (B): This could be a decision tree within the forest. The underlying process for completing the classification job for the RF is more complex. The authors either avoid such a presentation or make it more apparent in the figure title.

Yes, the underlying process is more complex and Supp. Figure 2B only showed the decision tree with the least number of nodes which could have been misleading for non-bioinformaticians. Thus, we agree with the reviewer to avoid such a presentation and the original **Supp. Figure 2** has been removed from the edited version of the manuscript.

– Supp. Figure 4: A 2D correlation plot would be used for the same purpose. A network presentation is not relevant here, especially from the ‘Network analysis’ viewpoint.

We agree with the reviewer that the original Suppl. Figure 4 is redundant, and is no longer in the manuscript. The 2D correlation plots are shown in **Figure 5A**.

Reviewer #2

The manuscript by Cabral-Marques et al. investigates the relationship between autoantibodies against GPCRs or RAS-related molecules and COVID-19 outcomes. In this cross-sectional study 169 SARS-CoV-2-infected individuals and 77 matched uninfected controls were compared for their autoantibody profile. Results showed that disease severity is significantly associated with highest autoantibody levels and that antibodies reacting against downstream molecules in the RAS cascade as well as CXCR3 have the strongest association with severe outcome.

The manuscript is well-written and relevantly contributes to deepen the knowledge of COVID-19 pathogenesis.

We appreciate the positive comments made by REVIEWER #2, which really helped us to improve our manuscript. Each comment is discussed point by point below.

I have some minor points:

-Due to the cross-sectional design of the study, it is unclear whether such autoantibodies were pre-existent or developed in response to SARS-CoV-2 infection. Were there any patients in the enrolled cohort who suffered from previous autoimmune diseases? And is any information available concerning the development of post-COVID autoimmune phenomena in the enrolled cohort? Did patients develop other “classical” autoantibodies during SARS-CoV-2 infection (e.g., ANA, aPL, ENA, RF, etc.?). That would be of crucial importance in connecting SARS-CoV-2 infection with the risk of full-blown autoimmune disorders in a more generalized virus-induced loss of self-tolerance. That would be of crucial importance in connecting SARS-CoV-2 infection with the risk of full-blown autoimmune disorders in a more generalized virus-induced loss of self-tolerance.

Thank you very much for this insightful comment, which is in agreement with recent reports of high levels of classical autoantibodies associated with autoimmune disorders in COVID-19 patients. We added the information about the “classical” autoantibodies in **manuscript lines 77-82**: “Additionally, we found that the dysregulated production of autoantibodies targeting GPCRs and the RAS in COVID-19 patients was accompanied by higher levels of some classical autoantibodies associated with autoimmune diseases³⁹ when compared to healthy controls. For instance, while we found no significant differences in antinuclear antibodies (ANAs), the levels of antibodies targeting double-stranded DNA (dsDNA) and rheumatoid factor (RF) significantly increased according COVID-19 severity (**Figure 2D**).” We also discuss the issues raised by the reviewer in the revised manuscript (**lines 199-222**) as follows: “The random forest model revealed an overlap between the autoantibody patterns of the moderate and severe COVID-19 groups, suggesting that an increase in autoantibody levels accompanies progression from mild disease. Our cross-sectional study cannot show whether these

antibodies were generated de novo. However, Chang et al⁵⁹ reported a subset of antibodies against autoantigens similar to those in classical autoimmune diseases as well as anti-cytokine antibodies that are generated de novo following SARS-CoV-2 infection. We have also identified higher levels of classical autoantibodies (anti-dsDNA and RF) in our COVID-19 cohort versus healthy controls and the details about their relationship with the clinical features of COVID-19 will be published elsewhere. Chang et al⁵⁹ also showed that while some autoantibodies were at or below the average levels of healthy controls and increased over time during the SARS-CoV-2 infection, other autoantibodies were already present in some seropositive patients, which is in accordance with the recently reported studies on preexisting autoantibodies to type I IFNs in COVID-19 patients^{60,61}. These data suggest that SARS-CoV-2 infection increases the production of autoantibodies or that at least some COVID-19 patients might have an unreported or undiagnosed autoimmune disease. To the best of our knowledge, none of our patients had previously experienced autoimmune diseases and we do not have follow-up information available concerning the development of post-COVID autoimmune phenomena in the enrolled cohort. In this context, since a previous report¹³ of our group indicates that anti-GPCR autoantibodies are natural components of human biology that can dysregulate and trigger the development of autoimmune diseases (a concept discussed in detail elsewhere)^{15,62}, we cannot exclude the possibility that at least some of our patients had dysregulated levels of autoantibodies targeting GPCRs and RAS prior to SARS-CoV-2 infection. Thus, a limitation of our report that needs further investigation is the lack of a longitudinal analysis of anti-GPCR/RAS antibodies to evaluate their levels from disease onset until convalescence. Further, any potential link to post-acute COVID-19 syndrome remains to be investigated.”

-Although demographics, therapeutics and clinical outcomes of the whole original cohort of SARS-CoV-2-infected patients have been described elsewhere (ref. 77,78), it would be useful to readers to report such data for the 169 selected patients and 77 controls in a table.

We fully agree that such data are important and need to be shown. We provided the demographics, therapeutics and clinical outcomes in manuscript **Supplementary Table 1**.

It seems that asymptomatic SARS-CoV-2-infected individuals were not included in the analysis. The authors should motivate the reason for this choice.

They were not included in this article because at the time of this study asymptomatic patients were not that common based on positive PCRs. Thus, without a corresponding PCR it would be difficult to establish that they were in fact asymptomatic COVID patients. Thus, “Of note, we did not include asymptomatic individuals in this article because their sera were not available at the time of data acquisition for this study. However, we are currently performing a follow up study with a German cohort of COVID-19 patients and have so far observed that healthy controls fully recovered from COVID-19 have a pattern of autoantibodies targeting GPCRs and RAS-related molecules that resembles

that from healthy controls and mild COVID-19 patients (manuscript in preparation).” This information is present in the revised **manuscript, lines 82-87**.

- Have gender, age, peripheral lymphocyte count (and relative lymphocyte subsets) and medications (e.g., steroids, antimalarials, immunosuppressive or biological agents) influenced the synthesis of these autoantibodies and to what extent?

We address this issue in the manuscript results. However, we do not have peripheral lymphocyte counts available to assess their association with the autoantibody levels. Kindly, see **manuscript lines 128-142**: “Of note, the aforementioned results were adjusted for age and sex in the production of autoantibodies by randomly selecting age- and sex-matched healthy controls and COVID-19 patients, reducing the likelihood of confounding effects. As a further precaution, we also assessed whether sex and age were associated with the top 10 autoantibodies ranked as predictors of disease severity by random forest analysis (**Supp. Figure 2A and 2B**). Overall, except for the MAS1-aab, which was significantly higher in control females versus control males, there were no sex differences in the COVID-19 disease groups. We also further analyzed whether the use of medications was associated with the levels of these autoantibodies and observed significant changes in the levels of some autoantibodies in severe patients receiving vitamin C and zinc (**Supp. Figure 3A-D**). However, this observation requires future investigation, because the influence of several other variables could not be controlled for in our study such as the inclusion of placebo as well as time and dose-response groups. In this context, it will also be important to assess the relationship between autoantibody levels and peripheral lymphocyte counts to evaluate, for instance, the impact of changes in the number of B lymphocytes on the serum levels of autoantibodies.”

- The discussion section provides several hypotheses for the pathogenicity of autoantibodies targeting the RAS and GPCRs withing other pro-inflammatory circuits in COVID-19, but no mention is made of neutralizing and binding antibodies that may indeed affect physiological pathways in a distinct way.

We included the hypothesis of autoantibodies targeting the RAS and GPCRs have neutralizing activity in the discussion. Kindly see **manuscript lines 250-255**: “Overall, although we postulate that dysregulated autoantibodies targeting GPCRs and RAS represent a pathological autoimmune phenomenon, it is also possible that some of them may have neutralizing activities, which requires future investigation. Considering the role of the immune system in homeostasis beyond host defense⁷⁴⁻⁷⁶, these autoantibodies could also represent both a physiological attempt of the immune system to promote body homeostasis during SARS-CoV-2 infection.”

- In the main section, please update the results by Wang et al. saying that both asymptomatic SARS-CoV-2-infected individuals (rather than healthy

controls) and COVID-19 patients have multiple antibodies against the exoproteome.

We updated the results by Wang et al. Please, see **manuscript lines 28-31**: “Wang et al.¹¹ showed that COVID-19 patients have multiple autoantibodies against the exoproteome. While patients with mild disease or asymptomatic infection exhibit increased autoantibody reactivity relative to uninfected individuals, those with severe disease have the highest reactivity scores.”

- In the discussion, the authors report that patients with moderate COVID-19 symptoms present with strong antibody production and high titers of neutralizing antibodies (ref. no. 50). This is only in part true. In their paper, Garcia-Beltran et al. rather write that “altogether, severity of SARS-CoV-2 infection significantly correlates with higher anti-RBD antibody levels, but suboptimal neutralization potency is a significant predictor of mortality”.

Thank you for this observation. This part pointed by the reviewer is no longer in the manuscript.

- In the patient cohort section, please provide a reference for the WHO severity classification of COVID-19.

The reference was included:

WHO. Clinical Management of Covid-19—Interim Guidance. Available online: <https://www.who.int/publications/i/item/clinical-management-of-covid-19>. 25 January 2021.

- Please note that figure 4 is erroneously miscalled as figure 5 in the legend.

We corrected this mistake

- Beside controls, Suppl. Table 1 reports a list of COVID-19 patients with mild, severe and oxygen-dependent disease but no patients with moderate disease. Please verify.

This was an old file mistakenly present in our supplementary material. It has been replaced by the correct one.

Reviewer #3

Key points:

This study by Cabral-Marques and colleagues draws on previous work from this group, notably the finding of autoantibodies targeting the G protein-coupled receptors (GPCRs), proposing that these autoantibodies are natural constituents of human biology that become dysregulated in autoimmune diseases. Given that Covid-19 infection often presents with immunologic features, these investigators sought to determine whether such autoantibodies were present, and what the clinical significance was

based on their presence and extended the work to autoantibodies targeting the immune and the renin-angiotensin systems (RAS) including GPCRs: MASR, AT1R, and AT2R in addition to ACE-II, which play an essential role in the development of severe COVID-19. Furthermore, the concentration of autoantibodies targeting GPCRs involved in chemotaxis, inflammation, and coagulation were studied, as well as neuronal receptor antibodies and antibodies targeting neuropilin, and STAB1.

Drawing on their previous reported that hierarchical clustering signatures of anti-GPCR autoantibody correlations are associated with physiological and pathological immune homeostasis, they investigated the autoantibody “correlation signatures” in healthy controls and patients with COVID-19 to determine if changes in autoantibody relationships correlate with disease severity.

We thank the Reviewer #3 for his/her positive comments on our manuscript. Indeed, we developed this manuscript based on the premise that anti-GPCRs are natural constituents of human biology that become dysregulated in autoimmune diseases. Thus, we carried out similar analyses to investigate the relationship between autoantibodies targeting GPCRs and the renin-angiotensin system and found that they associate with COVID-19 severity.

Insight:

The paper shows that autoantibody network signatures were relatively conserved in patients with mild COVID-19 compared to healthy controls, and such signatures were disordered in moderate and most perturbed in severe patients. Canonical-correlation analysis (CCA) was added to the Bivariate correlation analysis to try to better discern the autoantibody correlation signatures.

We acknowledge the reviewer to recognize the relevance of understand the autoantibody correlation signatures in COVID-19 patients in comparison to healthy controls.

Significance

The precise mechanisms by which SARS-CoV-2 infection prompts the production of autoantibodies remains unknown. When compared with patients manifesting mild disease, those with moderate COVID-19 symptoms demonstrate increased production of autoantibodies. The authors contend that their data show autoantibodies targeting GPCRs and RAS-related molecules are associated with COVID-19 burden. They assert that the association provides further evidence for the the proposed immunohematological mechanisms underlying the development of COVID-19 infection (which is grounded in the proposed abnormal activation of the ACE-II/Angiotensin II (Ang II)/AT1R/RAS axis together with a decrease of the ACE-II/ Angiotensin-(1-7)/MASR branch.

They suggest that their data provides new insights into the biology of

autoantibodies, which they contend is in line with their previous observation that “GPCR-specific autoantibody signatures associate with physiologic and pathologic immune homeostasis.”

They postulate several possible mechanism including the targeting of new epitopes during severe Covid-19.

Future provocative studies include whether the production of anti-GPCR autoantibodies is .

These are potentially significant and provocative findings and assertions; however, they rely on the accuracy and conclusions of the statistical modeling provided. Additionally, it is unclear how such modeling can be clinically useful unless an online calculator is provided, or there is another way to translate a “risk score” based on autoantibody correlations.

Some mechanistic explanations for the study’s findings are proposed but not yet proven.

We appreciate the comments from Reviewer #3 about the significance of our article. In accordance with his/her comments, we pointed out the limitations of our study in the discussion and comment along this section that the detailed mechanistic actions of the anti-GPCR autoantibodies remain to be investigated (e.g., **manuscript line 182-191**): “In this context, while the mechanistic action of several autoantibodies that we identified remains to be investigated, we previously described^{13,15,52,53} that anti-AGTR1⁵⁴ and anti-CXCR3 (previous work⁵⁵ and unpublished data) have agonist properties (e.g., on cell migration) and associate, for instance, with pulmonary fibrosis and cardiac death. Thus, these autoantibodies possibly potentialize the signaling triggered by their natural ligand, promoting the migration of immune cells, such as CD4+ and CD8+ T cells that are critical for both the killing of SARS-CoV-2 in the lung but also exacerbate deleterious hyperinflammation^{56,57}. Regardless that we did not investigate the activity of autoantibodies on their targets, the results of our work underscore those of recent studies^{3,6,10-12} that have reported the generation of autoantibodies following SARS-CoV-2 infection.”

Clarity and context:

The methods were appropriate, notably regarding patient selection and matching to controls. The text is clearly written and easily accessible with regard to the prose but the complicated mathematical modeling, while explained, is difficult to follow for non-statisticians or those with minimal biostatistical background training. .

We created a new figure (**Figure 1**; Study workflow) that outlines the study design in a concise form for the reader. The figure describes each set of analysis that we carried out to characterize the signature of autoantibodies targeting GPCRs and RAS-related molecules in COVID-19 patients in comparison to healthy controls.

References

The references appear to be appropriate and correct although this paper is predicated largely upon this group's prior work, and there is less consensus outside of their findings that actually verifies their prior findings, That is a troublesome aspect.

The fact is that we neglected to express the relationship of the present study with our group's prior work*. Therefore, we re-wrote the first paragraph of the manuscript introduction.

**Cabral-Marques, O. et al. GPCR-specific autoantibody signatures are associated with physiological and pathological immune homeostasis. Nat. Commun. 9, 5224 (2018).*

Display items

The tables and figures are nicely displayed but difficult for a non-statistician to follow. I think the authors attempt to walk the unseasoned reader through their findings and potential significance but in the absence of a biostatistical background, the data is hard to follow.

Kindly refer to the new **Figure 1A** (Study workflow) designed to provide guidance to the unseasoned reader. The figure describes each set of analysis that we carried out to characterize the signature of autoantibodies against GPCRs and COVID-19-associated molecules in COVID-19 (the renin-angiotensin system) patients in comparison to healthy controls.

Suggested improvements

I have limited suggestions for the authors as I think the manuscript reads well; however, it relies on the prior findings of the group as well as a biostatistical analysis that may be generally inaccessible to most readers. Reproducibility by other investigators would also lend more credibility to these findings.

Please, refer to the above response. All data generated for this study are provided as supplementary material. In addition, all R packages used in this manuscript are now available at the link: <https://github.com/lSchimke/The-relationship-between-autoantibodies-targeting-GPCRs-and-the-renin-angiotensin-system-associates->

Thus, any other investigator can reproduce the biostatistical analysis of our manuscript.

Minor suggestions:

(1) This group has previously reported on autoantibodies targeting the G protein-coupled receptors (GPCRs) suggesting that these autoantibodies are natural components of human biology that become dysregulated in autoimmune diseases. This information should be incorporated into the abstract for context as to what these were ascertained in Covid-19 patients.

Thank you for this comment. We edited the manuscript abstract and incorporated the information that autoantibodies targeting GPCRs are natural components of human biology that become dysregulated in autoimmune diseases. We believe that this suggestion really improved the abstract.

(2) Last paragraph under MAIN section: “However, these investigations were not systemic” – I think the authors mean to say systematic. If not, they need to clarify what they mean by “systemic” here.

We restructured the last paragraph of the main section (**manuscript lines 38-45**) to clarify that “However, these investigations focused only on a few anti-GPCR autoantibodies. Importantly, they did not investigate their relationship with the potential presence of autoantibodies targeting molecules of the immune and renin-angiotensin (RAS) systems, which play a central role in the development of severe COVID-19. Thus, we employed a systems immunology approach (**Figure 1A**) to characterize the relationship between autoantibodies targeting a broad group of GPCRs and the RAS with COVID-19 severity by determining their correlation signatures across SARS-CoV-2 infected patients versus healthy individuals.”

(3) Third paragraph under DISCUSSION section: “Furthermore, our work indicates a change in the relationship between autoantibodies targeting GPCRs and RAS that associate with COVID-19 severity, which was shown by increasing disruption of autoantibody correlations according disease burden.” The word “to” is missing before the word disease.

The word “to” was included before the world disease in **manuscript line 239**.

Expertise:

The statistical analyses here are outside my scope of expertise and I strongly suggest that a biostatistician verify the analyses and claims of the significance of the results.

We included a schematic **Figure 1A** of the study workflow to improve the clarity of the manuscript.

REVIEWERS' COMMENTS

Reviewer #1 (Remarks to the Author):

All my previous comments have been addressed nicely.

Reviewer #2 (Remarks to the Author):

The authors have fully addressed my comments.

I only recommend verifying the correctness of Suppl. Table 1, since it still shows patients with mild, severe and oxygen-requiring disease but not those with moderate disease included in the analysis, and of reference 81 (please refer to <https://apps.who.int/iris/bitstream/handle/10665/338882/WHO-2019-nCoV-clinical-2021.1-eng.pdf>).

Best Regards

Reviewer #3 (Remarks to the Author):

Thank you for addressing the points of concern that I raised. All of my concerns were addressed adequately.

Point-by-point response

January 17th, 2022.

REVIEWERS' COMMENTS

Reviewer #1 (Remarks to the Author):

All my previous comments have been addressed nicely.

We thank Reviewer 1 for the critical review of our manuscript and appreciate that there are no more suggestions from his/her side.

Reviewer #2 (Remarks to the Author):

The authors have fully addressed my comments. I only recommend verifying the correctness of Suppl. Table 1, since it still shows patients with mild, severe and oxygen-requiring disease but not those with moderate disease included in the analysis, and of reference 81 (please refer to <https://apps.who.int/iris/bitstream/handle/10665/338882/WHO-2019->

nCoV-clinical-2021.1-eng.pdf).

We thank Reviewer 2 for his/her careful revision of our manuscript. Thank you very much for this correction, we checked the Supplementary Table 1 and corrected the classification of patients as mild, moderate, and severe COVID-19.

Reviewer #3 (Remarks to the Author):

Thank you for addressing the points of concern that I raised. All of my concerns were addressed adequately.

We thank Reviewer 3 for his/her important review and appreciate that all concerns were addressed adequately.